# Biopolymer-Based Wound Dressings with Biochemical Cues for Cell-Instructive Wound Repair

**DOI:** 10.3390/polym14245371

**Published:** 2022-12-08

**Authors:** Variksha Singh, Thashree Marimuthu, Maya M. Makatini, Yahya E. Choonara

**Affiliations:** 1Wits Advanced Drug Delivery Platform Research Unit, Department of Pharmacy and Pharmacology, School of Therapeutic Sciences, Faculty of Health Sciences, University of the Witwatersrand, 7 York Road, Parktown, Johannesburg 2193, South Africa; 2Molecular Sciences Institute, School of Chemistry, University of the Witwatersrand, Private Bag 3, PO WITS, Johannesburg 2050, South Africa

**Keywords:** cell instructive, biochemical cues, biomaterials, biomimetic, biopolymers, wound healing

## Abstract

Regenerative medicine is an active research sphere that focuses on the repair, regeneration, and replacement of damaged tissues and organs. A plethora of innovative wound dressings and skin substitutes have been developed to treat cutaneous wounds and are aimed at reducing the length or need for a hospital stay. The inception of biomaterials with the ability to interact with cells and direct them toward desired lineages has brought about innovative designs in wound healing and tissue engineering. This cellular engagement is achieved by cell cues that can be biochemical or biophysical in nature. In effect, these cues seep into innate repair pathways, cause downstream cell behaviours and, ultimately, lead to advantageous healing. This review will focus on biomolecules with encoded biomimetic, instructive prompts that elicit desired cellular domino effects to achieve advanced wound repair. The wound healing dressings covered in this review are based on functionalized biopolymeric materials. While both biophysical and biochemical cues are vital for advanced wound healing applications, focus will be placed on biochemical cues and in vivo or clinical trial applications. The biochemical cues aforementioned will include peptide therapy, collagen matrices, cell-based therapy, decellularized matrices, platelet-rich plasma, and biometals.

## 1. Introduction

Emerging smart therapies are needed to mitigate the challenges posed by chronic wounds. Bioactive therapies actively influence tissue responses. In contrast, biomimetic therapies guide tissue responses inspired by the extracellular matrix (ECM). Designing wound healing dressings based on the latter approach of supplanting cellular strategies to utilize the body’s repair code offers an innovative avenue in wound management strategies.

Wound repair is a biological process that varies with regard to the entirety and the period of time to attain healing [1]. Thus, wounds can be characterised as either acute or chronic. The former conforms to the normal physiological progression of the wound healing mechanism, while the latter deter from the usual repair pathway. These wounds pose great difficulty when it comes to healing and is known to exceed 12 weeks to resolution. Chronic wounds arise when the healing process arrests in one of the phases and can be attributed to many factors such as microbial infection, poor blood supply, underlying pathological conditions (venous or arterial disease), sustained irritation, wound dehiscence, etc. [2]. Delayed healing can also be attributed to the expression of matrix metalloproteinases (MMPs) and other proteases produced by inflammatory cells. They prevent the secretion of growth factors (GFs) and cytokines that are crucial in the normal healing process. Chronic wounds, such as diabetic foot ulcers, venous leg ulcers, and pressure ulcers, are complex and can become infected, subsequently making clinical practice an imperative component of wound management [3]. The chronic wound environment contains high levels of proteases that cause extracellular matrix (ECM) breakdown and the inhibition of cell proliferation and migration. They are also in a constant state of inflammation due to bacterial toxins, thus posing the risk of abscess formation, cellulitis and/or limb loss [4]. Chronic wounds are pervasive in the high-risk population such as geriatrics, immunocompromised individuals, obese patients, and many more. In addition, the rapid growth of diabetic occurrence leads to an even greater emergence of chronic wounds.

Wound healing physiology is primarily controlled by cellular cues. The cellular microenvironment presents cells with stimuli that co-ordinate an array of cellular activities [5]. The direction provided by microenvironmental cues is responsible for the trajectory of cells and physiological endpoints. Tissue engineers have exploited this knowledge to develop biomimetic, instructive platforms to usurp the body’s innate repair mechanisms [6,7]. For instance, as seen in Figure 1, growth factors and cytokines have well-known roles in controlling cellular behaviours; therefore, by recounting their biochemical signals within a biomaterial, the system can enhance natural healing pathways.

For adequate wound closure, a highly efficient biomaterial is needed. It should promote an optimal microenvironment for epidermal regeneration and barricade the wound bed from moisture loss and infection [9]. Traditional gauzes and dressings are mostly dry and can adhere to the wound surface, causing pain and discomfort upon dressing changes. The new generation of biomaterials in the form of electrospun nanofiber mats, hydrogels, and nanocomposites possess unique properties suitable for accelerating wound healing [10].

Formulation of the most suitable polymer and bioactive for the composition of a wound healing platform is a vital component in bringing about attuned wound healing [11]. Polymer choice also affects the physiochemical aspects of the gel, such as its thermos-responsiveness or pH-sensitivity to the wound environment [12]. When considering hydrogels for wound therapy, the mechanical strength and bioactivity of the gel are important factors. Therefore, polymers that impart these properties alongside biocompatibility are desirable in formulation design.

Biomaterials that encapsulate cellular cues have gained heightened interest in mitigating the growing insurgence of chronic wounds. Owing to the attractive nature of biopolymers, this review will focus on biopolymeric wound dressings encoded with instructive biochemical cues for wound healing applications. This review will therefore highlight the interplay between the biomolecule and its encapsulation into polymeric-based hydrogels and or scaffolds to provide a perspective on how future optimised systems for wound healing can be developed.

We will, firstly, outline the phases of wound healing with a spotlight on the major influencers; secondly, effective biopolymers employed for wound healing will be explored; lastly, insights into wound dressings embedded with instructive biochemical prompts will be provided.

## 2. The Phases of Wound Healing

Wound repair is an intricate physiological process that occurs to restore tissue integrity upon injury. Following an acute injury, a series of four overlapping steps occur in a highly orchestrated manner, as seen in Figure 2. Coagulation and haemostasis occur primarily to stop blood loss. Vasoconstriction occurs, followed by a platelet plug formation and an insoluble fibrin mesh reinforcement [13]. Scab formation is present in the finality of this stage for the purpose of blood clotting and protecting the wound microenvironment from external pathogens.

Secondly, inflammation occurs wherein immune cells infiltrate the scene, and numerous inflammatory cytokines, proteases, endopeptidases, and reactive oxygen species (ROS) are also present [14]. Neutrophils and macrophages are responsible for debridement. Neutrophils are removed from the site of injury in the form of slough and eschar. Monocytes differentiate into specialised macrophages that remove microorganisms and apoptotic cells by phagocytosis. Macrophages house growth factors, such as transforming growth factor-β (TGF-β) and epidermal growth factor (EGF), which play a regulatory role in the inflammatory stage [15]. In the later stages, macrophages also stimulate angiogenesis and granulation tissue formation. To prepare for the third proliferative phase, macrophages shift from a pro-inflammatory (M1 phenotype) state to an anti-inflammatory, pro-regenerative state (M2 phenotype) [16].

The proliferation stage begins in concession once the inflammatory response is balanced and the debris is eliminated. Angiogenesis occurs as a vascularization technique [17]. Cellular mediators such as epithelial cells, fibroblasts, keratinocytes, stem cells, and M2 macrophages intrude into the wound bed for repair. Herein, they lay down ECM proteins (hyaluronan, fibronectins, and proteoglycans) and produce collagens [18,19]. The collagen acts as a temporary scaffold for granulation tissue formation [20]. They impart strength to the infant tissues. The wound defect is filled with granulation tissue, and epithelialisation occurs. Epithelial cells actively divide and migrate inwards to form a continuous layer of cells to repopulate the wound bed. Wound contraction is carried out by myofibroblasts. Growth factor release, angiogenesis, epithelialization, keratinization and collagen deposition are all key features of the proliferative phase.

Finally, the regeneration stage takes place, during which the functionality of the assaulted area is reinstated to its pre-injured form. Remodelling occurs as the deposited collagens, and other proteins evolve into organized matrices. Collagen undergoes remodelling from type three to type 1, thus increasing the tensile strength and completing wound healing. This physiological process is heavily directed by collagens and relies on its innate ability to support and strengthen as regeneration occurs. Skin appendages are also reformed. In most instances of serious wounding, remodelling does not occur or reach completeness due to fibrosis dysregulation. Therefore, healing results in scarring with some loss of tissue capacity.

**Figure 2 polymers-14-05371-f002:**
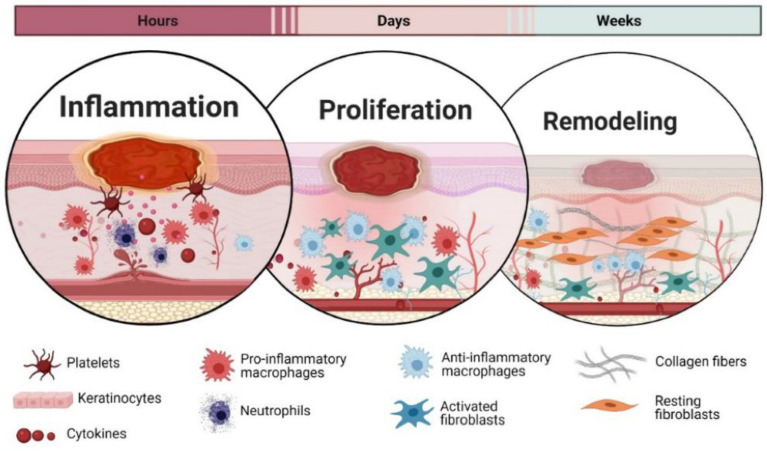
The main stages of wound healing show the roles of the various cell types and triggers. Reproduced from Petkovic et al. [21].

## 3. Functional Biopolymer-Based Wound Dressings

Biopolymers in wound healing are sought after due to their versatile qualities such as mechanical strength, biocompatibility, biodegradability, non-toxicity, non-immunogenicity, and biomimicry of the native ECM [22,23,24,25]. Furthermore, biopolymers can be formulated to enable physical interactions for desired functionalities or chemically modified to give rise to specific responsive systems. Tissue engineers make use of biopolymers to produce artificial polymeric dressings. Biopolymer-based wound dressings are advantageous in comparison to traditional dressings as they induce rapid healing, maintain a moist wound climate, possess antimicrobial functionalities, and ensure maximal contouring of the wound bed to reduce pain and allow for efficient healing [26]. Biomaterials can also be employed as delivery vehicles for functional bioactive molecules [27]. In the next section, leading natural biopolymers for wound healing are reviewed, and their biological properties will be highlighted. The structure of some of the polymers central to wound repair can be seen in Figure 3 and include chitin and chitosan, starch, sodium alginate, cellulose, carrageenan, pectin, and hyaluronic acid.

### 3.1. Chitin and Chitosan

Chitin and chitosan are the most prevalent polysaccharides. Chitosan is a deacetylated derivative of chitin to enhance the water solubility of the molecule. Chitin can be extracted from the shell food waste of crustaceans [28]. Both chitin and chitosan possess remarkable properties such as biocompatibility, biodegradability, antimicrobial activity against gram-positive and negative bacteria, haemostasis, and good adhesion to cells [29]. As a result, they play a crucial role in wound healing. These biopolymers have been shown to facilitate wound repair, neovascularization, and dermal regeneration and show analgesic, antimicrobial, and haemostatic properties [30]. Chitosan has the ability to form complex, multilayered structured hydrogels for enhanced results in the fields of drug delivery, tissue engineering, and cell culturing [31]. Moreover, drug molecules can be loaded onto chitin/chitosan materials for enhanced therapeutic effects [32]. Chitosan and its derivatives have been formulated in numerous drug delivery platforms. Li et al., formulated multifunctional haemostatic microbeads using carboxymethyl chitosan and hyaluronic acid, cross-linked with starch and tannic acid [33]. The resultant dressing showed a 3D network for cell growth, antibacterial effects, and rapid haemostasis. Wound repair was seen within 14 days. Leonhardt et al., composed chitosan hydrogels loaded with β-cyclodextrin polyester (CDPE) analysed in vivo on many different animal models consisting of rats, rabbits, and pigs. This system showed superior haemostatic properties in comparison to commercially available dressings [34].

### 3.2. Starch

Starch is an abundantly found natural biopolymer composed of anhydrous glucose units that form different polymers. The first being amylose and the second being amylopectin. Upon heating, starch undergoes gelatinization, during which the granules swell and release amylopectin [35]. The by-product serves numerous roles in wound healing [36,37]. Moreover, starch is colourless and transparent in nature, allowing for wound monitoring without having to remove the dressing. Mao et al., developed a hydrogel from oxidized starch and gelatin as non-invasive medical suture for wound closure [38]. It showed self-contracting properties and thermos-responsive shape memory due to polymer reactions. The hydrogel showed successful wound repair and facilitated tissue reconstruction in an in-vivo rabbit model. It also displayed smoother skin and less visible scarring in comparison to typical sutures.

### 3.3. Alginate

Alginate is extracted from marine brown algae, which is an extremely common and renewable source [39]. Upon the addition of ionic crosslinking agents, alginates tend to swell and form hydrogels. Gelation is possible due to the interaction between the glucuronic-rich portions of adjacent polymer chains with the subsequent formation of an “egg-box” shaped structure [39]. Alginates are readily used as wound dressings as they maintain an optimal wet wound environment and reduce the presence of bacterial microbes [40]. Due to their high absorption capacity, alginates are ideal for moderate-excessively draining wounds. Moreover, these polysaccharides activate macrophages to produce inflammatory cytokines, thus, rapidly enhancing wound-healing processes [41]. Alginates famously stimulate monocytes for the production of IL-6 and TNF-α that have a direct role in the healing process [42]. Lu et al., developed a multifunctional self-healing alginate hydrogel that delivered a sustained release of strontium ions to the wound bed [43]. This formulation was evaluated on rats and showed rapid angiogenesis and accelerated wound repair. Its self-healing ability promoted the hydrogel for clinical applications that required exercises.

### 3.4. Cellulose

Cellulose is an important polysaccharide that has a vital place in wound repair therapies. It is found in bountiful quantities and is easily obtained from different sources, such as natural cellulose, bacterial cellulose, and plant cellulose. Carboxymethyl cellulose hydrogels are known for wound exudate absorption, flexibility, inducing angiogenesis, and autolytic debridement [44]. Cellulose reaps its rewards in wound healing applications due to its moisture-retaining, exudate-absorbing properties, and porous structure, producing ECM mimicry. Liu et al., developed a physically crosslinked polydopamine/nanocellulose pH-responsive hydrogel [45]. The hydrogel exhibited sustained drug release and good antibacterial and mechanical properties.

### 3.5. Carrageenan

Carrageenans (CRGs) are polysaccharides extracted from red seaweeds, Rhodophyceae. They have a high molecular weight and are anionic, linear, and sulphated in nature (Figure 3). This group can be subdivided into six categories based on their sulphate concentration, extraction techniques, and solubility. Of the six groups, Kappa (κ)-, Iota (ɩ)-, Lambda (λ) are most sought out owing to their viscoelastic and gelling abilities [46]. Carrageenans structurally resemble naturally occurring glycosaminoglycans and are ductile, which is a promising feature regarding skin contact [47]. Carrageenans have been used extensively in drug delivery, and food packaging and are now promising candidates in tissue engineering and regenerative medicine [46,48]. A limiting factor for wound healing relates to its inflammatory and immunomodulatory properties [46]. Tavakoli et al., developed a nanocomposite bioadhesive hydrogel that was crosslinked by light curing [49]. It was composed of methacrylated kappa-carrageenan, polydopamine-modified ZnO nanoparticles, and L-glutamic acid and showed multifunctional properties. Physicomechanical properties of the crosslinked hydrogels included mechanical strength and recovery, comparable to the elasticity of human skin and excellent bioadhesiveness. In vivo studies revealed accelerated wound healing with superior granulation tissue thickness.

### 3.6. Pectin

Pectin is a simple heteropolysaccharide found in a plant’s cell wall comprising galacturonic acid with minimal amounts of neutral sugars in its side chain [50]. Pectin has been exploited for its anticancer, antimicrobial, wound healing, drug carrier, anti-oxidant, and anti-inflammatory actions [51]. Giusto et al., formulated an economically advantageous pectin-honey hydrogel that showed accelerated wound healing in rat models in vivo [52].

### 3.7. Hyaluronic Acid (HA)

Hyaluronan is a glycosaminoglycan prominent in the ECM [47]. Its high molecular weight HA shows anti-inflammatory abilities, whereas its low molecular weight equivalent shows proinflammatory activities. Much evidence supports the role of HA in the maintenance of tissue integrity, dermal regeneration, cell adhesion during the inflammatory stage, and improving wound repair mechanisms [53]. HA is influential in modifying tissue injury, accelerating tissue repair, and determining the consequence of a wound [54]. HA is reputable for its ability to promote angiogenesis, granulation of tissue, and remodelling of the ECM [55]. Moreover, HA has shown active control over cell proliferation and differentiation during wound healing. Da Silva et al., developed a spongy HA hydrogel impregnated with stem cells [56]. The formulation enhanced wound closure rates (Figure 4a,b), positively impacted re-epithelialization, and regulated inflammatory responses (Figure 4c,d), leading to a successful neoinnervation of diabetic mice full-thickness wounds. Huang et al., formulated an injectable chitosan/hyaluronic acid hydrogel with porous Poly (lactic-*co*-glycolic acid) microspheres loaded with VEGF and vancomycin [57]. This hybrid system exhibited antibacterial effects and accelerated endothelial cell proliferation in vitro. In rat models, the hydrogel was effective in managing non-healing infected wounds by modulating inflammation and endorsing angiogenesis.

### 3.8. Collagen

Collagen is revered as a fundamental protein in the extracellular matrix (ECM), bone, and ligaments. Fibrous collagens provide mechanical strength, while network-like collagens make up the structural integrity of the basement membrane that ultimately organizes tissues [58,59]. The stability of its structure and ability for tailoring has made it a forefront candidate for scaffolding in tissue engineering. Collagen offers physical and biological features that include haemostatic abilities, enhancement of neo-collagen laydown, and management of proteolytic activity in chronic wounds. Collagen also plays a critical role in blood clotting and cascade initiation. Collagen-based wound dressings are beneficial as they increase the production of fibroblasts and fibronectin, and preserve leukocytes, macrophages, fibroblasts, and epithelial cells [20]. Overall, they maintain an optimum microenvironment of a wound to enhance and quicken the healing process. Zheng et al., developed a novel bio-inspired collagen aggregate/chitin-based “cotton-like” biomimetic 3D microstructure [60]. The formulation possessed haemostatic abilities, promoted cell proliferation, migration, and accelerated wound healing.

### 3.9. Gelatin

Gelatin is a polypeptide with a higher molecular weight owing to it being derived from partially hydrolysed collagen [61]. For the finest healing, collagen synthesis and breakdown needs to be balanced. The breakdown of collagens by MMPs brings rise to gelatin. In the wound milieu, gelatin provides a moist environment, enhances angiogenesis, scab disintegration, and improves growth factor-cell contact [62]. Gelatin has proven to influence wound healing by augmenting the homeostasis stage and absorbing wound exudate. Jeong et al., evaluated a complex coacervate (EGF-Coa) consisting of low molecular weight gelatin type A and sodium alginate [63]. The formulation reduced pro-inflammatory cytokine release, enhanced in vitro migration of keratinocytes and accelerated wound healing in diabetic mice by amplifying granulation tissue formation and re-epithelialization.

### 3.10. Silk

Silk has been used in wound management in the form of sutures for many centuries. Silk fibroin proteins are derived from silkworm cocoons. This protein acts by integrin-mediated cell adhesion due to the presence of the tripeptide (Asp(*R*)-Gly(G)-Arg(D); thus, enabling cell proliferation and migration [64]. Attractive features of silk fibroin include its ability to absorb exudate, good permeability, offer mechanical strength, and wound healing action by inducing neovascularization, enhanced re-epithelialization, and tissue ingrowth [64]. Liu et al., developed a neurotensin/gelatin microspheres/silk fibroin dressing for treating full-thickness wounds [65]. The formulation accommodated fibroblast release, collagen deposition, and organisation, as well as controlled release of neurotensin for DFU treatment.

Silk sericin is a natural material from the cocoon of the domestic silkworm *Bombyx Mori*, procured from the waste of the silk industry. It is composed of 18 amino acids with broad moieties, namely -OH and –COOH. This biomedical substance is renowned for moisture retention, antioxidant, anti-bacterial, anti-tyrosinase, anticoagulant, anticarcinogenic, and UV resistant [66,67]. Gilotra et al., developed a nanofibrous matrix-based wound dressing using poly(vinyl alcohol) incorporated with silk sericin [68]. The dressing demonstrated good in vivo performance in terms of absorption of fluids, and antibacterial and antioxidant activity.

### 3.11. Keratin

Keratin consists of insoluble cysteine-containing proteins. Keratin extracted from human hair has shown promise in skin regeneration as dermal substitute [69]. Keratin has been developed into films, hydrogels, dressings, and scaffolds intended for use in a variety of biomedical applications, which include nerve regeneration, wound repair, tissue regeneration, and cell culture [70,71]. Keratin confers both structural and bioactivity as well as good adhesive properties [72]. Ye et al., fabricated a keratin-based nanofibrous membrane loaded with silver nanoparticles [73]. The formulation displayed favourable antibacterial properties, good biocompatibility, and improved wound closure in vivo.

While biopolymers are subjects of interest in the wound community, there are several disadvantages to their place in biomaterial manufacture. These include limited shelf lives, batch-to-batch variability, and inadequate mechanical properties [22]. However, through appropriate processing and novel combinations, complex biopolymer-based archetypes that offer a tunable structure, comparable to the native ECM, can be obtained for clinical practice. The abovementioned common biopolymers, chitosan, alginate, collagen, cellulose, and hyaluronic acid, are summarised in Table 1.

## 4. Biochemical Cues in Wound Healing Applications

### 4.1. Peptides as Pioneering Materials for Wound Healing Applications

Many studies have evaluated the impregnation of biochemical cues in scaffolds. These molecules often exert their effect by adhesion and binding to ECM, other cells, etc., through specific binding receptors. There is potential here for substitution with synthetic peptides that can, via molecular engineering, mimic these adhesion sequences [84]. By achieving this, synthetic peptides can control cell fate through adhesion-mediated signaling, which, in turn, influences cell migration, proliferation, and differentiation [84,85].

#### 4.1.1. Antimicrobial Peptides

Chronic wounds develop due to poor debridement and microbial infiltration by opportunistic pathogens such as *Staphylococcus aureus* and *Pseudomonas aeruginosa*, thus delaying proliferation and regeneration. In addition, upon colonisation of the non-healing wound by microorganisms, biofilm formation can occur, making the wound even more difficult to manage [86].

Important host-defence molecules include antimicrobial peptides (AMPs). These molecules are produced in response to the infiltration by potentially harmful pathogens. AMPs are composed of amino acids of varying lengths; these molecules are oligopeptides. Most AMPs are cationic in nature. Antimicrobial activity is brought about by the electrostatic interactions with the anionic phospholipids of the microbial membrane, thus inducing cell death.

LL37, a human cathelicidin, is reputable for its antibacterial, antifungal, antiviral, and antiparasitic activity [87]. It exerts effects during the early phases of wound healing. It has been shown to result in the production of inflammatory cytokines such as interleukin-6 (IL-6), TNF-β, and interleukin-1β (IL-1β) through the stimulation of keratinocytes [88]. It also sets about inflammation and can therefore eliminate microorganisms. LL37 proved to have a dual action of a pro-inflammatory molecule as well as an anti-inflammatory molecule. For this reason, the AMP has been dubbed “immunomodulatory” in nature. LL37 additionally showed angiogenic properties and keratinocyte migration. Simonetti et al., evaluated his peptide in methicillin-resistant *Staphylococcus aureus* (MRSA)-infected wounds in vivo [89]. Apart from confirming its bactericidal activity, results showed the value of this peptide to wound repair. Upon topical administration, wound closure was enhanced in this study by the stimulation of granulation tissue formation as a result of well-organized collagen lay down and epithelium reconstitution. Authors highlighted that LL37 induced the migratory isomer of keratinocytes through the mediation of the EGFR pathway. In addition to keratinocyte-boost, LL37 exhibited angiogenic capacities by promoting the release of vascular endothelial growth factor (VEGF) and neovascularization through the direct association with G-protein-coupled formyl peptide receptor-like 1 (FPRL-1).

AMPs have faced a struggle in their translation to clinical use due to numerous factors [90]. The evidence of cytotoxicity has capped their use in topical applications. They also exhibit poor stability under physiological conditions. In attempts to circumvent toxicity concerns of AMPs, Annabi et al., evaluated sprayable, elastic UV-crosslinked composite hydrogel composed of GelMa/ methacryloyl-substituted recombinant human tropoelastin (MeTro) and the antimicrobial peptide Tet213 (KRWWKWWRRC) [91]. Favourable biocompatibility and biodegradability were established in a rat model with good potential for the treatment of chronic non-healing wounds. Further, Li et al., investigated an in situ forming hydrogel consisting of poly (L-lactic acid)-Pluronic L35 nanoparticles loaded with antimicrobial peptides 57 (AP-57) [92]. The reported thermoresponsive gel showed a prolonged release of AP-57. In vitro, the formulation showed low cytotoxicity and high anti-oxidant properties. When tested in vivo on full-thickness models in Sprague Dawley rats, the gel significantly promoted wound healing. The optimal AMPs have been identified as those derived from nature. It has been proposed that these peptides can be used in synergism with other therapies like mesenchymal stem cell-based (MSC) and platelet-rich plasma therapies (PRP) to overcome the challenges experienced clinically and promote coordinated biomodulation in tissue engineering [87]. Wei et al., acted upon this theory to develop a composite hyaluronic acid hydrogel for the co-delivery of AMPs and PRP in-vivo infected diabetic wounds [93]. They developed a composite hydrogel that was formulated using oxidised dextran (ODEX), and AMP-modified hyaluronic acid (HA-AMP) incorporated with PRP. In vitro, the PRP hydrogel facilitated enhanced fibroblast proliferation and migration. The AMP/PRP hydrogel showed accelerated wound closure, denser collagen deposition, lower bacterial load, and regulated inflammation. It exhibited a reduction in inflammatory responses by downregulating IL-1β, TNF-α, IL-6, and TGF-β secretion. Additionally, through the upregulation of VEGF, the gel was responsible for the enhanced expression of cluster of differentiation 31 (CD31) and α-Smooth muscle actin (α-SMA). Ultimately, it was the collagen deposition and VEGF-mediated angiogenesis that resulted in wound repair.

#### 4.1.2. Collagen Mimetic Peptides

The importance of collagen in physiological processes and its limited natural form have led to the development of short synthetic peptides modelled from collagen, known as collagen-mimetic peptides (CMPs) [94]. These short peptides are endowed with the collagen triple helical motif that comprises repetitive glycine (Gly)-X-Y tripeptide units, where X and Y are usually proline (Pro) and Hydroxyproline (Hyp), respectively. Since their inception, many research groups have attempted to modify the molecule by substituting units in the side chain, terminus, and backbone [95,96,97]. Additionally, CMPs are capable of forming higher-order assemblies via nucleation and growth mechanisms. Refractory works have manipulated CMP side chains to form coordination complexes with metal ions and ligands for controlled self-assembly of CMPs [98]. Furthermore, CMPs are able to bind to innate collagen via a strand hybridization process [99]. CMPs anneal to damaged collagen strands and anchor bioactives to a wound bed, as seen in Figure 5 [100]. This propensity of CMPs to bind to native collagen brought about the ability to encode cellular cues into the molecule and act as the bioactive domain [101], as well as to label and trace the molecule in vivo [102,103]. CMPs in hydrogels have demonstrated their ability to act as physical cross-linkers in additional to containing instructive chemical cues that enhance cell adhesion, proliferation, and ECM production. Moreover, CMPs can form conjugates with polymers to produce hybrid materials that show promise as targeted drug carriers [104]. These hybrid materials are capable of forming self-assembled nanostructures at physiological temperatures. CMPs have been successfully conjugated to elastin-like peptides (ELPs), yielding thermos-responsive, smart biomaterials intended for collagen-associated diseases [105].

##### Collagen Mimetic Peptides with Integrin Targeting Motifs

Integrins are transmembranous receptors that orchestrate cell-ECM and cell-cell interactions. Integrins permit connection and communication between the various cell types, including parenchymal and non-parenchymal cells. Thereby, integrins are keystones in the commencement and trajectory of cells to a healed state.

Integrins are a collection of heterodimeric receptors that are studded across the cell membrane. Each receptor is composed of one α and one β subunit and is formed from the association of 18 α and 8 β. There are 24 different integrin receptors, with each specifically binding to one or more ligands. Integrins provide a link between the inside and outside of a cell. These receptors intercept two signals, i.e., intracellular to extracellular space and vice versa. It is through this action that integrins control the architecture of the cytoskeleton. As a result, they influence cellular commands such as adhesion, migration, proliferation, survival, and differentiation. Furthermore, integrins are responsible for detecting changes to the extracellular matrix in times of pathological disturbances such as wound healing, fibrosis, and cancer. Integrins then cause cellular responses that control ECM remodelling. Integrins are capable of binding ECM components, as well as, binding cells to other cells via A Disintegrins (ADAMs) and matrix metalloproteinases, to encourage matrix remodelling, as well as vascular cell adhesion molecules (VCAMs) and intracellular adhesion molecules (ICAMs) expressed on endothelial cells and leukocytes, respectively.

Integrin-mediated cell adhesion to collagen is a critical physiological process that is linked to embryogenesis, haemostasis, tissue remodelling, and healing [107]. Integrin receptor binding sites on type 1 collagen are responsible for binding to numerous cell types, such as platelets, chondrocytes, fibroblasts, and osteoblasts [108]. Collagen is abundantly located in the ECM. Several hexapeptide sequences GXX’GER on natural collagen have been identified as integrin binding sites such as GLOGER, GMOGER, GASGER, GROGER, with a particular spotlight on GFOGER moiety. These motifs partake in integrin binding, thus imparting a functional role on collagen. Once bound, it is these bioactive motifs that trigger and domino cytoplasmic intracellular signalling pathways. CMPs are able to be modified to allow for biofunctional activity. For instance, GFOGER derived from residues 502–507 of collagen alpha 1, is a sequence specific to integrins and has shown to be a major cell binding motif with collagen type 1.

Growth-factor derived therapies aimed at augmenting wound repair are prominent in pre-clinical trial stages but lack translation due to a lack of efficacy. As a solution, GF gene therapies have emerged. These are aimed at regulating multiple aspects of cellular behaviours via the controlled expression of extracellular cues resulting in a wealth of localized GFs in the wound bed to promote repair. Gene-based approaches allow the host cell to orchestrate a sustained GF expression, with a subsequent increase in efficacy. Gene-encoding therapies in conjunction with platforms have a synergistic effect in wound healing; however, they are lacking due to off-target and immune responses, inefficient gene transfer in serum-dense environments, and inability to produce effects for the duration of the healing period. Urello et al., adopted an approach of using CMPs to overcome the counterpart issues associated with gene delivery [109]. They exploited the ability of CMPs to bind to collagen and the ease of tailoring CMP strand sequences. Their findings showed that CMP-modified DNA-PEI polyplexes had the ability to improve control over the extent and duration of gene expression. Additionally, in serum-rich environments, CMP modification allowed for maintained activity. Interestingly, CMP-modified polyplexes had the capability to “hijack” collagen remodelling. Through the use of collagen remodelling for gene release, the CMP-modified agent has the ability to enhance polyplex activity due to an increase in the preservation of DNA integrity in serum-dense environments and an increase in caveolar uptake. In this study, a CMP-modified DNA-polyethylenimine (PEI) polyplex encoded with platelet-derived growth factor-BB (PDGF-BB) was nominated as the test molecule. The CMP [(GPP)3GPRGEKGERGPR(GPP)3] was synthesised by Fmoc solid phase methods and purified using reverse-phase HPLC. It showed enhanced polyplex activity over extended periods in simulated wound environments and increased PDGF-BB levels. Significant increases in cell counts were exhibited in the cell proliferation studies for the CMP-modified polyplexes compared to the unmodified polyplexes. The study also showed improved contraction and enhanced migratory effects in vitro. When evaluated, in vitro, 3D wound models, wound closure was at 90% in 10 days compared to the 40% closure exhibited by the unmodified polyplex scaffolds. These findings speak to the utility of this agent in actual wound repair.

Thappa et al., investigated the treatment of chronically infected wounds (Thapa et al., 2020) [110]. *Staphylococcus* and *Pseudomonas* are the most common micro-organisms that populate chronic wounds to bring about infection. Their approach was to formulate a topical application as an alternate to oral antibiotics that would localise and control the level of active drug to maximise wound healing effects. To achieve this, they adopted the use of hydrogel impregnated with liposomes to facilitate controlled release. Particle retention posed an issue due to the high fluid base of chronic wounds with a subsequent lack of absorption and rapid wash-off. This issue was overcome by the use of CMPs. The group synthesised a 35 amino acid CMP [(GPP)3GPRGEKGERGPR(GPP)3GPCCG] via an automated Fmoc solid-phase peptide method. This study conjugated CMPs to the surface of vancomycin-loaded liposomes (CMP-Van-Lipo) (Figure 6a). These particles were subsequently integrated into collagen scaffolds to retain the particles allowing for the sustained release of vancomycin. The tethered CMP-Van-Lipo co-gels showed a significantly greater sustained release of vancomycin at 48 h, in comparison with Van-Liposomes and Van-Lipo co-gels. Additionally, this system had greater antibacterial effects in vitro. The formulation was tested in vivo in an MRSA-inoculated murine excisional wound. Punch biopsy technique was utilized, with the wounds formed inoculated with MRSA. Herein, it showed promise for antibacterial activity. Histological analysis was performed and presented with better epidermal layer formation (Figure 6b), comparable to the non-infected wound models and minimal/no bacterial colonies. This study coherently highlighted its potential as a topical antimicrobial sustain release treatment for the control of MRSA-infected chronic wounds.

### 4.2. Collagen Matrices Embedded with Biochemical Cues for the Promotion of Wound Healing

Collagen is notorious for being the most abundant protein in mammals. In wound healing, it has a determinant role in the remodelling phase to impart tensile strength and support scaffolds. Despite the numerous collagen types, all collagens display a unifying repeating Gly-X-Y motif, where X and Y can be any amino acid but are most commonly proline and 4-hydroxyproline, respectively. This unit is responsible for forming the characteristic right-handed triple helical conformation of collagen. Secondary structures of collagens form through self-assembly into fibrils, subject to post-translational modifications. While collagen plays a structural role, it is also vital in relaying biochemical signals to the cells. Processes such as cell adhesion, migration, proliferation, and metabolism are all directed and modulated by prompts supplied by collagen. Pathologies arise when there are deviations in the collagen types that are expressed.

Collagen in wound therapy exerts its effect by acting as a decoy for the MMPs and other forms of enzymatic degradation; a potent inducer of cellular migration; and an advocate of a proangiogenic, anti-inflammatory wound setting [111].

Collagens are widely used in drug delivery, cell culture, and tissue engineering due to their gelling properties and biodegradability [81]. Collagen type 1 has widespread use in tissue engineering as it is easily sourced and fabricated into scaffolds that present physiological cues to regulate cellular behaviour [112]. Collagen scaffolds are attractive since they can provide cells with the biochemical and biostructural cues that are intrinsic to natural collagen. They contain adhesive peptide motifs that bind to integrins. Additionally, they encompass peptide sequences that are targeted and degraded by MMPs, and they present fibres that topographically guide cues to the cells [59,113,114]. However, the use of animal-derived pure collagens poses restrictions in terms of its difficulty in tuning and chemically modifying environmental degradation (pH, temperature, etc.), batch variation, immunogenicity, and pathogenic contamination. As research evolves, scaffolds with biomimetic landscapes and a high capability for tuning are sought after. There is a paradigm shift from pristine collagen matrices to engineered collagen mimetic environments. The resulting outcome is to engineer scaffolds that pageant the important constructs of collagen while allowing for enhanced modifications.

Engineering matrices from collagen-derived materials is an emerging technique that is built on the use of pure collagen matrices. There are numerous strategies to achieve this outcome [115]. Firstly, gelatin-based matrices are formed from the thermal denaturing of collagen and form a polymer with an almost identical amino acid composition to collagen. Moreover, there has been significant growth in the use of methacrylated gelatin (GelMA) as a scaffold material. Berger et al., reported on the application of a UV light polymerised (Figure 7A) interpenetrating network (IPN) of gelatin-methacrylate and collagen that allowed for tuneable fibre density and scaffold stiffness (Figure 7B) [116]. Secondly, collagen mimetics, as a synthetic approach, are applied to mimic specific biochemical and biophysical topographies of collagen. As mentioned above, this approach is made possible through the development of synthetic collagen-mimetic peptides comprised of repeating Gly-X-Y units to yield triple helical structures. It is also possible to synthesize self-assembling fibres that mimic the design of natural collagen without the Gly-X-Y motif. Lastly, and interestingly, another tactic for constructing collagen-mimetic environments is to reduce collagen to its elemental bioactive. This technique involves the incorporation of precise bioactive collagen-based peptide arrangements into a scaffold that displays collagen-specific bioactivity. Collagen is an adhesive protein and contains the RGD peptide motif for this purpose. The peptide sequence GFOGER (Gly-Phe-OHPro-Gly-Glu-Arg) and DGEA (Asp-Gly-Glu-Ala) are also dominant adhesive moieties in collagen 1 [115]. Mimicking these sequences in a scaffold allows for the adhesion of cells to the ECM to promote reciprocal healing. Alternatively, a peptide motif for matrix metalloproteinase recognition can be embedded into a scaffold. A popular peptide sequence for this is GPQGIAGQ [115].

Chattopadhay et al., anchored a cytoactive factor, substance P, to the wound bed through the use of a collagen-mimetic matrix [106,117]. A CMP was used as a pylon and bound to damaged collagens but not intact triple helical collagens. In this way, it localized the cytoactive factor to the damaged wound bed. Upon in vivo evaluation of splinted wounds in db/db mice, enhanced wound healing was seen, with extensive re-epithelialization and regulated inflammatory responses. In this scenario, the collagen matrix provides structural support by annealing to damaged collagen, as well as introducing cytoactive agents to promote healing.

### 4.3. Cells as Directing Prompts for Enhanced Wound Healing

Cell-based therapies emerged to meet the need for more complex and biomimetic therapies with moisture-retentive capacities and clinical relevance [118]. Cell-based therapies aim to bring balance to dysregulated inter-cellular communication pathways. This group has been explored in many dressing forms, such as biopolymeric scaffolds, electrospun fibrous meshes, hydrogels, and 3D-bioprinted matrices (Heras et al., 2022) [118].

Cellular therapies have shown great potential for future use in wound repair technologies with >136 registered clinical trials. This category consists of many subcategories, such as immune cells, bone marrow mononuclear cells, leukocytes, progenitor cells, and stem cells. Of these, the most promising are mesenchymal stem cells (MSCs) derived from bone marrow, adipose tissue, umbilical cords, and the placenta. MSCs show potential due to their immunomodulatory and pro-regenerative characters [119]. MSCs have a paracrine ability that can realign dysregulated systems in the wound healing micro-cascade, thus bringing about ripple effects in re-epithelialization, angiogenesis, granulation tissue deposition, and preventing apoptosis of wound healing factors [120,121]. These effects are arbitrated by growth factors (angiopoietin 1 (ANGPT1), EGF, HGF, FGF, KGF, insulin-like growth factor-1 (IGF1), platelet-derived growth factor (PDGF), VEGF, TGF-β, immunoregulatory factors (interleukin-10 (IL-10)), chemokines (C-C motif ligand 2 (CCL2), antimicrobial peptides (LL-37), stromal-derived factor 1(SDF-1)) as well as exosomes and micro-vesicles that house pro-regenerative factors [119,121]. The effects of MSCs and mediators are seen in Figure 8.

Bone marrows derived MSCs are broadly recognised as the foremost cell supplier and therefore have been the attention of many animal studies. Nilforoushzadeh et al., developed MSC spheroids inoculated in an injectable thermoresponsive hydrogel [122]. MSC 3D spheroids were implanted into alginate microbeads with a subsequent impregnation of the beads into a hydrogel that gels at skin temperature. In vivo studies on rabbit ears, full-thickness wounds showed granulation and re-epithelialisation after a week. Wound regeneration was accredited by the MSC encroaching upon the body’s repair mechanism to secrete high levels of cytokines IL-10 and TGFβ1. Moreover, the expression of α-SMA was amplified, leading to wound contraction after two weeks. The quality of wound repair was also improved, and this can be attributed to the steady decline in α-SMA secretion. In conjugation with this affect, reduced scar formation was also aided by neovascularization and parallel fine collagen bundles.

Systemic cell delivery posed problems due to low cell engraftment at the wound site. As a result, other delivery platforms have been investigated, such as scaffold-abetted cell grafting. Ribeiro et al., exhibited the use of human MSCs derived from the umbilical cord Wharton’s jelly with polyvinyl alcohol hydrogel membrane in two dogs presenting with non-healing large skin lesions [123]. After treatment, both animals showed significant skin regeneration with a lessened degree of ulcerated lesions. Additionally, Navone et al., developed an electrospun nanofibrous silk fibroin scaffold cellularised with human adipose-derived mesenchymal stromal cells for diabetic wound healing [124]. This formulation was tested in vivo on mice and was found to reduce the wound area by 40% within three days. This group showed superior healing through the expression of epidermal and collagen fibres being well organized, with new hair follicles present in comparison to the controls. In-vitro ELISA analysis showed a significant increase in angiogenic molecules such as FGF2, TGF-β, and EGF. Tissues were analysed in vivo for gene expression, which showed a significant upregulation of numerous genes associated with angiogenesis, ECM deposition, and remodelling. This system pays dividends through the release of factors that promote angiogenesis and collagen deposition.

The use of MSCs faces the challenge of procurement, preparation, characterization methods, ideal doses, and delivery platforms. New science has emerged around the use of the secretome of MSCs, harvested in vitro upon the direction of environmental cues, to create cell-free therapeutics that optimize the release of trophic factors in exosomes [125]. Apart from animal studies, cell therapies have been evaluated in clinical trials. Moon et al., studied the potential of hydrogel-based allogenic adipose-derived stem cells (ASC) sheets for treating diabetic foot ulcers (DFU) in a randomised clinical trial [126]. A total of thirty patients were treated with the ASC sheets, while twenty-nine patients were treated with a control polyurethane film. The ASC group exhibited enhanced wound closure (82%) compared to the control group (53%) over a period of twelve weeks. There was no evidence of rejection of ASC treatment. In another study, Sarasu’a et al., examined autologous bone marrow mononuclear cells containing MSCs and other stem/progenitor cells, for the treatment of patients with pressure ulcers secondary to spinal cord injuries [127]. In this study, 21 patients with single-type IV pressure ulcers greater than four months in duration were treated with BM-MNCs. Notably, nineteen patients showed fully healed ulcers after 21 days with no recurring ulceration within a nineteen-month tracking period.

### 4.4. Decellularized Matrices as Regenerative Biomaterials for Wound Healing

The ECM is an integrated latticework comprising protein and polysaccharides that provide physical support and stores and delivers a cocktail of GF and cytokines. Moreover, the ECM is involved in cell adhesion, migration, proliferation survival, and differentiation (Rasouli et al., 2021) [128]. The stem cell niche is a concept describing the specific cell milieu that relays extracellular cues for stem cell survival and differentiation. It has been proven that ECM proteins are critical components for the inception of the stem cell niche (Rasouli et al., 2021) [128]. The resulting outcome is that any dysregulation in the secretion or degradation of ECM constituents will cause ripple effects on cell and tissue function. Harmonious regulation of the ECM is, therefore, necessary for tissue efficiency. If cell attachment to the ECM is defective, the interplay between the cell and its ECM is prohibited, thus leading to anoikis.

When a wound is incurred, chemotaxis occurs, thereby directing progenitor cells to regulate the proliferative healing phase (Dash et al., 2018) [129]. To that end, changes in the ECM can contribute to the re-direction of progenitor cells to tangent from the path of repair. Matricellular proteins include tenascin (TN), osteopontin (OPN), CCN and SPARC, and thrombospondin (TSP). They are a colony of proteins responsible for cell-matrix interactions. They bind to cytokines, cell surface receptors, and proteases. In effect, they are critical factors in cell proliferation and migration as cytokines and chemokines attach to these proteins (Rasouli et al., 2021; Volk et al., 2013) [19,128]. Matricellular proteins contribute to ECM lay-down and are upregulated during ECM remodelling. Activated platelets secrete matrix constituents, including matricellular proteins. TSP-1 is a matricellular protein engaged in regulating cell-matrix interactions, activating TGF-β, and upregulating the expression of fibrin and collagen [130]. Another matricellular protein is tenascin-c, which actively influences re-epithelialization, filopodia production in fibroblasts for migration, fibronectin deposition and the prevention of premature matrix contractions before adequate collagen laydown. Osteopontin affects collagen remodelling. Galectin-1 aids healing and fibrosis regulation by preventing the activity of vitronectin.

Matrix elements communicate with progenitor cells through integrin receptors. Integrins are transmembranous receptors responsible for cell adhesion and interactions. Integrins are the link between the cytoskeleton and the extracellular components such as laminin, fibronectin, elastin, and collagen fibres [131]. The vital role of ECM-stem cell interactions in wound healing is mediated by ECM components and stem cell integrins (Figure 9). Stem cell self-renewal, migration, differentiation, and anoikis are influenced by the ECM. These processes are integrin-mediated bi-directional signalling, i.e., inside-out signalling allowing for integrin interactions with ECM fibres and outside-in signalling [128]. Integrin expression in the epidermis causes stem cells to interact with the ECM, thus accelerating healing. Integrins also play a role in the differentiation of myofibroblasts, thus controlling wound contraction to complete repair and reducing the incidence of scars. Additionally, integrins contribute to neovascularization and granulation tissue lay-down as they are expressed on endothelial cells.

As previously highlighted, MSCs are useful strategies in wound healing therapies as they are able to infringe upon cellular mechanisms. However, evidence suggests that these progenitor cells become vulnerable in physiological conditions due to cell death, hostile environments, and inadequate ECM supplies for cell-ECM interactions [132]. To that end, using ECM matrices for MSC delivery is justified for biomimicry. When tissue is decellularized, it is termed acellular ECM. Through processing, hydrogels can be harvested to form ECM-based systems for clinical applications [133]. The ECM then promotes cell-ECM interactions and encourages cell signalling, adhesion, migration, and proliferation in wound healing. The shortcomings of this approach include host-cell rejection [134]. An ongoing challenge is to maintain a balance between extracting sufficient cellular materials to prevent immune responses while still protecting the integrity of the ECM composition to induce reciprocal actions for repair. Huang et al., developed an acellular dermal matrix seeded with ASCs for the treatment of full-thickness cutaneous wounds in murine models [135]. In the treatment group, granulation thickness was enhanced. After nine days of the operation, an increase in re-epithelialization rate for the acellular dermal matrix (ADM)-ASC group compared to the control group was noted, thus enhancing wound healing. In addition, the blood vessel density was also increased on day 14, thus contributing to new vasculature and angiogenesis. In another study using an acellular dermal matrix pretreated with MSCs, Sahin et al., evaluated their formulation on murine models [136]. They combined their development with negative pressure wound therapy (NPWT). It was revealed that ADM-MSCs with combined NPWT presented with increased graft adherence. Their findings suggested that MSC transplantation included angiogenesis to a greater extent than NPWT. A combination of these therapies was proposed to work in synergism to improve angiogenesis and neovascularization.

There is a need for standardised decellularisation conventions that preserve key tissue-specific features and produce systems with established cell-instructive effects.

### 4.5. Platelet-Rich Plasma in Progressive Wound Healing Applications

Platelet-rich plasma (PRP) strategies have been projected as a unique approach in regenerative medicine to kick-start the tissue repair cascade [137]. There are a number of currently approved PRP therapies. Herberprot-P^®^ (intralesional application) and Regen-D™ 150 (gel form) are rhEGF-based commercially available products for the treatment of DFU. Easyef^®^ is another PRP-GF-based dermal spray intended for DFU. The Aurix™ system and SafeBlood^®^ are commercially available autologous systems suitable for at-home preparation and immediate application for wound healing [138]. Platelets have a keystone role in the initial phases of wound repair. This is achieved by its autologous nature and granule release during activation in the heamolytic plug, which triggers the body’s own healing actions [138]. It is through this granule release that platelets exert their effect on wound repair by promoting coagulation and releasing an assortment of growth factors (GF’s), chemokines, and cytokines. Subsequently, their actions are reaped in angiogenesis, proliferation, and differentiation of injured cells [138]. During heamostasis, platelets are tasked with the formation of the platelet plug to avert blood loss. Upon inflammation initiation, platelets induce chemotaxis to bring in neutrophils which then commence phagocytosis of bacteria [139]. Platelet factor CXCL4 also contributes to the chemotaxis of neutrophils and macrophages and regulates degranulation [140]. Additionally, platelets activate complement factors, engage with interleukins and stimulate TGF-β to recruit immune response cells [141]. Platelets act in the proliferative phase via the secretion of GFs (TGF-β and PDGF) to induce fibroblast migration and the subsequent fibroblast-mediated release of proteoglycans, hyaluronan, fibronectin, and collagen to synthesize matrix [140]. Proceeding from this, angiogenesis is aided by TGF-α, FGF, TGF-β, VEGF, and PDGF [141]. During remodelling, tissue re-organization and deposition are highly regulated by MMPs. Platelets perform in this stage by collagen deposition, ECM proteins, and augmenting MMPs by inducing the downstream upregulation of tissue adhesion proteins and tissue inhibitors of metalloproteinase (TIMPs) [142]. Ultimately, these pathways lead to the recoiling of the matrix and wound closure.

PRP is a concentrated platelet preparation aimed at a cost-effective delivery platform for repair-orientated proteins. It is obtained from the patient’s whole blood, and plasma is extracted after centrifugation. The clinical shortcomings of GFs include their limited half-life, rapid degradation, low stability, and intravenous toxicity [138]. In attempts to overcome the adversities, delivery systems have been evaluated. The systems in questions must protect the GF from protease degradation, enhance retention, and minimize the high dosage and frequency of administration to achieve therapeutic effects. Hydrogels show promise as GF carriers since their capabilities include controlled release, cell adhesion, and miming the ECM [143]. Zhang et al., engineered a PRP-based dual-network sodium alginate hydrogel and tested it in vivo on rat models [144]. In vivo studies exhibited improved angiogenesis, faster re-epithelialization, and enhanced GF expression. Liao et al., evaluated an allogenic PRP therapy in the context of a refractory chronic wound in a prospective, randomized, single-centre study from January 2014–January 2018 [145]. After a month of treatment, the allogenic PRP-treated group exhibited pink granulation with lesser inflammatory exudate. The overall rate of healing for this group was accelerated in comparison to the control. Mohammadi et al., evaluated a PRP gel in a single-arm clinical trial for the treatment of DFU [146]. The results exhibited a diminution in the wound-ulcerated area by 51.9%.

### 4.6. Delivery of Biometals for Wound Healing Applications

Bioactive glasses are amorphous solid materials with a short-range order formed from oxides. The first bioactive glass was intended for bone regeneration, but once the doorways for experimentation were opened, it was discovered that they showed great potential in soft tissue engineering as well [147]. Their composition is founded on silicate, borate, and phosphate with the ability of bioactive incorporation, such as Ag, B, Ca, Ce, Co, Cu, Ga, Mg, Se, Sr, and Zn [148]. In wound healing applications, it is the release of therapeutic ions that target the various stages of healing to bring about the superior repair. They have revealed a high ability to induce angiogenesis and compel cellular functions and antibacterial mechanisms [148,149]. Periodic elements play a significant role in enzymatic reactions, cell functions, homeostasis of pH balance, and fluid regulation [148]. In a physiological setting, bioglass materials dissolve to accommodate the sustained and localized release of therapeutic ions. Via these means, optimum concentration windows are reached without veering to toxicity. Wound repair is promoted by the microstructure, composition, and ionic dissolution of glasses [150]. Bioactive glasses prompt the secretion of GFs (PDGF, FGF, TGF-β, VEGF), angiogenesis, haemostasis, antibacterial effects, cell viability, proliferation, and migration [148]. 45S5 is a common bioglass composed of four oxides, namely SiO_2,_ CaO, Na_2_O, P_2_O_5_. Silicon (Si) stimulates pro-angiogenic growth factors, improves skin elasticity and strength, and contributes to collagen synthesis and re-organization [151]. Secondly, calcium (Ca) plays a role in platelet plug formation, fibroblast proliferation regulation, collagen formation, keratinocyte proliferation, and differentiation [152]. Phosphorous (P) played a pivotal part in cellular signalling and proangiogenic factors (VEGF-α) through the display of phosphate ions.

Incorporating bioactive glasses into polymer platforms, such as ointments, polymeric matrices, or microfibrous dressings, capitalizes on the therapeutic properties of both these components, as seen in Figure 10. Bioglasses improve the functional ability and mechanical nature of the system, while the polymer matrix controls the release of the bioglass to prevent toxicity. Zeng et al., designed a composite injectable hydrogel ornamented with 45S5 bioglass/agarose-alginate (Zeng et al., 2015) [153]. The Bioglass-Agarose alginate (BG/AA) hydrogel is gelled upon physiological introduction. It created a moist wound environment and improved angiogenesis. The gel was sculpted by intercalation between the alginate and agarose. This cross-linking was further enhanced by calcium ion release from the bioglass interacting with alginate. In vitro studies were conducted on a human dermal fibroblast (HDF) and human umbilical vein endothelial cell (HUVEC) co-culture system. Here, the bioglass compelled the upregulation of VEGF gene expression, thus implying an angiogenic potential of the bioglass. Additionally, in vitro scratch assay findings showed an increase in cell migration of HUVEC and HDF upon hydrogel treatment. When conducted on animal studies in a rabbit ear wound, neovascularization was seen in amplified quantities compared to the control group. Another study, conducted by Zhou et al., appraised an injecTable 45S5 glass/albumin hydrogel that releases bioactive Si and Ca ions (Zhou et al., 2018) [154]. In this system, the glass works in symbiosis with human serum albumin (HSA) to form a gel. The glass dissolves to encourage an alkaline environment that accommodates the pH-mediated cross-linking reaction between succinimidyl succinate modified-polyethylene glycol (PEG) and the free amino groups of the HSA. This system displayed superior handling properties and adhesive abilities. Interestingly, they found an inversely proportional relationship between the bioglass quantity and the gelation time. Apart from the ability of the bioglass to regulate gel formation, it also contributed to the promotion of wound healing by endorsing angiogenesis. In-vivo studies further elaborated on this system’s ability in wound healing. It was shown to enhance collagen deposition, and stimulate the differentiation of fibroblasts to myofibroblasts to allow for wound contraction, keratinocyte migration for re-epithelialization, and angiogenesis. On day 7, the treated group exhibited thicker epidermal and dermal layers in comparison to the control group. The wound dressings encoded with biochemical cues are summarised in Table 2.

## 5. Further Perspectives

While bioinstructive cellular therapies are gaining traction is wound healing and tissue regeneration, there is room for improvement and optimization. As mentioned in this review, there are biochemical and biophysical cell cues that infringe upon regenerative mechanisms to propagate wound healing. We have extensively evaluated biochemical cues as they are more involved in wound healing. However, on the other end of the spectrum, there are biophysical cues. Biostructural cues are powerful mediators of cell biology. There are two major cellular components, namely adherens junctions, and focal adhesions, that assume responsibility for sensing and reacting to the biophysical microenvironment, thus, employing them as critical mediators in cell migration, differentiation, and ECM composition [157]. Understanding these systems allows for improved targeted platform design. Biostructural cues relate to stiffness, porosity, topography, fluid flow as well as stress relaxation [158]. To this end, synergistic strategies appear as a method to enhance the role of bioinstructive matrices [159,160]. This can be attained by combining biochemical and biophysical cues and mutually exclusively combining multiple biochemical or biophysical cues. Moreover, cellular cues can be combined with drug compounds for enhanced biomaterial development. Combining biochemical and biostructural cues form a superior, compounded biomaterial that can revolutionize the regenerative medical field.

Yao et al., developed a 3D bioprinted specific sweat gland (SG) like matrix by the differentiation of MSCs into functional SG that facilitated SG recovery in mice [156]. This was achieved by combining biochemical and structural cues. An alginate/gelatin hydrogel was used as the bio-ink. A 3D cellular macroporous construct, embedded with evenly distributed fibres, of cross cross-sections 30 mm × 30 mm and height of 3 mm was fabricated. 3D bioprinting provides an alternative delivery system for tissue repair and regeneration. These 3D constructs are seen to be biomimetic as they offer biochemical and biophysical signals for cell-cell and cell-ECM interactions. Their matrix capitalized the collagen triple helix chemical cue (CTHRC1) and the physical structure provided by the 3D construct. These two cues guided MSCs into a glandular lineage and functional SG recovery in vivo.

Bioinstructive molecules are aimed at controlling a cellular input and its complement behaviour/response. The cues in question can be achieved by recapitulating well-known biochemical motivators in the cell milieu aimed at unlocking the body’s innate mechanisms of repair. This improved mechanistic approach has guided tissue engineering to develop systems carrying cell instructive cargo zoned in on controlling downstream cell behaviour and improving tissue regeneration. In wound healing, this approach has been successfully adopted and executed in animal studies and clinical applications. Moreover, biopolymers actively influence wound healing, thus, the selection of the biopolymer/s during synthesis is crucial in outlining biological performance. Biopolymers also offer biophysical structural prompts to direct cells for advantageous healing.

Peptides as alternate materials for wound healing offer alternative approaches to the field of wound healing. GFs and extracellular proteins lack translation to biomedical applications due to environmental factors (oxidation, hydrolysis, photolysis) and wound-related issues (proteolysis and alkaline pH environments) resulting in low residence times. Owing to this, replacing them with their respective peptide binding sequences is an attractive alternative. To compound this revolutionary idea, bioactive peptides can be mass-produced at lower costs and limit non-specific binding. In this respect, CMPs and AMPs measure up to expectations and are intended to contribute to regenerative medicine.

Collagen matrices are attractive attributes in advanced wound healing. Reproducing the complex geometrical and architectural components of collagen is a forefront method in the fabrication of engineered collagen matrices. There is a plentiful indication of the prominence of biochemical attributes of collagen-mimetic strategies but also the biophysical collagen architecture that guides cell function.

Cell therapy for wound healing is a significant contributor to wound healing. Stem cells hold a noteworthy ranking in regenerative medicine due to it long-term self-renewal capabilities. Particularly in wound healing, stem cells directly or indirectly compel inhabiting cells, trigger the release of biomolecules, regulate inflammation and remodel with ECM. Once MSCs are supplanted into the cellular workings of wound healing, they regulate inflammation and fibrosis, promote fibroblast function, encourage angiogenesis and granulation tissue formation, reduce scarring, exert antimicrobial effects, stimulate endothelial progenitor cells and recruit MSCs to the injury site. Additionally, the ECM components play crucial roles in prompting stem cell fate and calibre of tissue repair. The ECM provides a locale for stem cells and allows for a dynamic bidirectional interplay with them. Stem cells remodel the ECM while their fate is subjected to the ECM, as well. A major strategy of tissue engineering encompasses the use of stem cell-speckled scaffolds on injured tissues for regenerative and restorative purposes.

ECM-derived matrices with pendant stem cells are utilised to release MSCs into the wound bed, infiltrate the cellular flow and bring about the faultless repair.

Platelet-rich plasma therapy is a promising candidate in wound healing. PRP is a concentrate of human platelets in a small volume of plasma. PRP contains various bioactive proteins, interleukins, and GFs that induce various cellular and biological functions such as cell proliferation, differentiation, and tissue remodelling. The clinical relevance of PRP is dependent on delivery platforms to offer mechanical strength, stabilise, and facilitate controlled release of the GFs. PRP holds value as a safe and economical therapy that is easily prepared and activated for use. An untapped sector of PRP therapy is personalizing the preparation through formulation strategies for individual patient use.

Ion therapy/bioactive glass holds promise as a wound-healing application. Bioactive glass and its dissolution properties exert biological impacts on cell processes to bring about wound repair. Additionally, some of their characteristics, such as haemostatic and antimicrobial properties, are advantageous in wound healing.

## 6. Conclusions

The bioinstructive avenue opens up intriguing possibilities for microenvironmental manipulations of cell fate. Tissue engineers have applied the understanding of the microenvironment to build biomimetic, instructive materials for wound healing applications. Recapitulating biochemical cues within a wound healing system allows for the direction of cells to advantageous ends. The challenge of understanding the signals and input instructions that are necessary to control cell behaviour is ongoing. Biophysical aspects of the cellular microenvironment also require in-depth comprehension. Substantial progression in the field of wound healing requires an understanding of how cells sense, interpret and respond to biochemical and physical prompts. Further development, based on the importance of biopolymers in wound healing systems, is required to manufacture effective designs. Improved understanding can guide wound healing-based applications to specifically target the necessary structures/pathways for improved repair and regeneration.

## Figures and Tables

**Figure 1 polymers-14-05371-f001:**
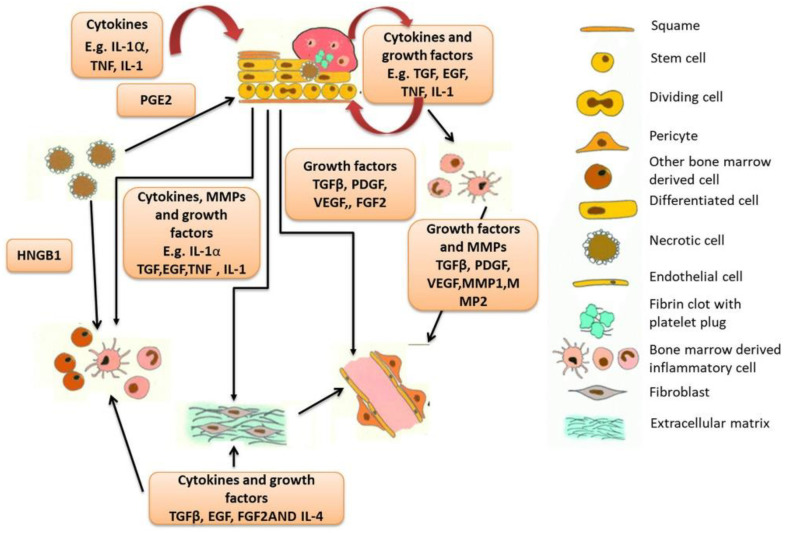
Growth factors employed in the wound healing process. Reproduced with permission from Sharma et al. [8].

**Figure 3 polymers-14-05371-f003:**
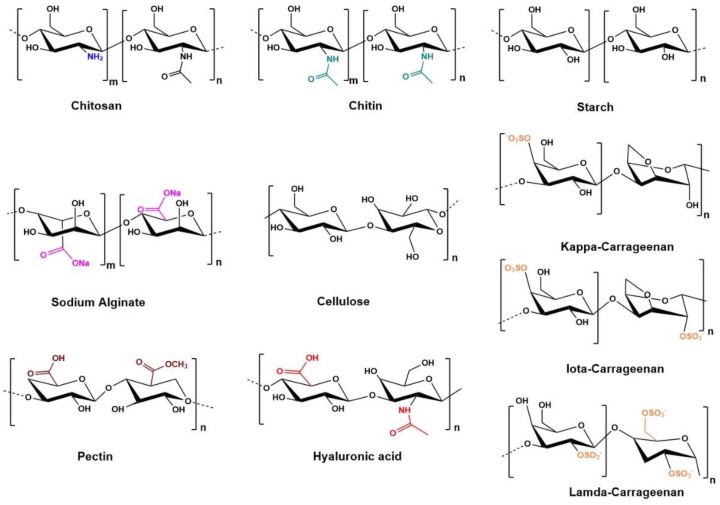
Functional carbohydrate biopolysaccharides highlighting their bioactive moieties.

**Figure 4 polymers-14-05371-f004:**
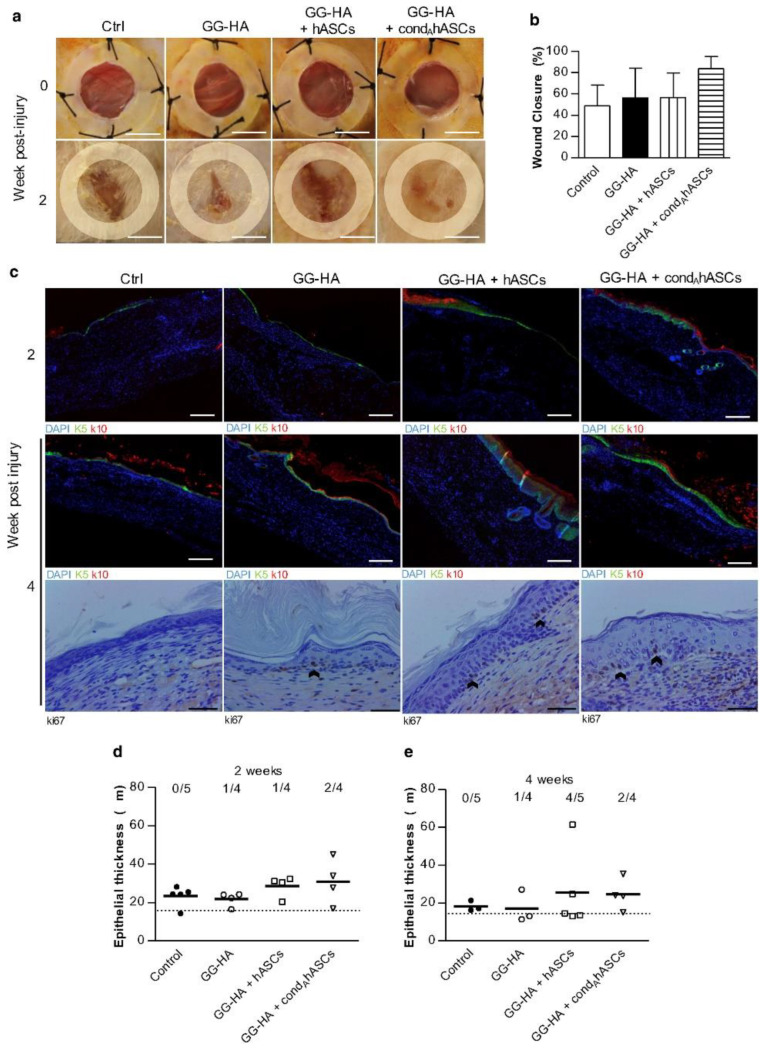
Wound closure and re-epithelialization 2 and 4 weeks post-transplantation with treatments in diabetic mice full-thickness wounds. (**a**) Photographs of the wounds and (**b**) the corresponding wound closure represented as a percentage. (**c**) Immunohistochemistry imaging. Quantification of the epidermal thickness after (**d**) 2 weeks and (**e**) 4 weeks. Where, cond_A_hASCs: human adipose-derived stem cells conditioned to neurogenic medium A; and GG-HA: gellan gum-hyaluronic acid. Reproduced with permission from Da Silva et al. [56].

**Figure 5 polymers-14-05371-f005:**
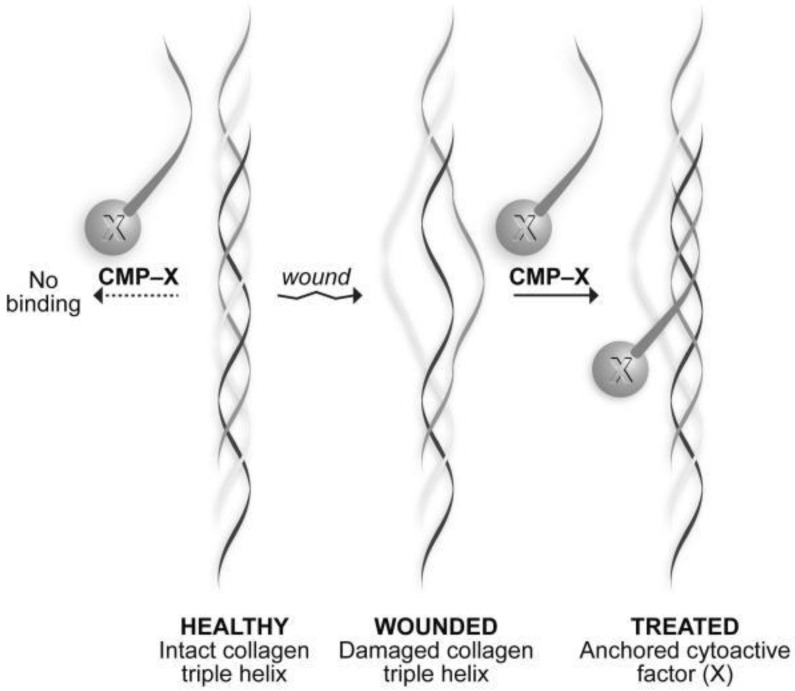
Depiction of a collagen-mimetic conjugate (CMP-X) annealed to the triple helix of a damaged collagen fibril. “X” represents the cytoactive that becomes indented in the wound microenvironment. Reproduced with permission from Chattopadhyay et al. [106].

**Figure 6 polymers-14-05371-f006:**
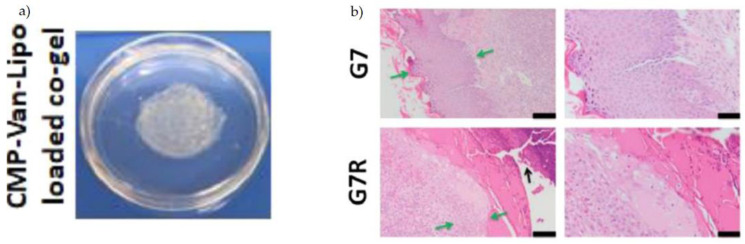
(**a**) Photograph of the CMP-Van-Lipo-loaded co-gel formulation and (**b**) Histological analysis performed on GR and G7 containing the test formulation of CMP-Van-Lipo co-gel. Where, G7: Punch biopsy wounds + bacterial inoculation + CMP-Van-Lipo loaded co-gel; G7R: Punch biopsy wounds + bacterial inoculation + CMP-Van-Lipo loaded co-gel + re-inoculation at day 1. Green arrows indicate epidermal thickness; black arrow indicates the presence of a clot. Adapted with permission from Thapa et al. [110].

**Figure 7 polymers-14-05371-f007:**
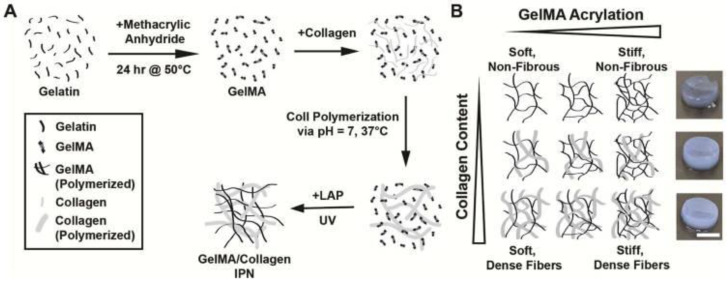
The formulation process of the gelMa and collagen IPN. (**A**) Collagen was polymerized in the presence of gelMa before UV-crosslinking in the presence of the photoinitiator LAP. (**B**) The varying amounts of collagen and degree of gelMa methacrylation allowed for independent tuning of fiber density and scaffold stiffness. Reproduced with permission from Berger et al. [116].

**Figure 8 polymers-14-05371-f008:**
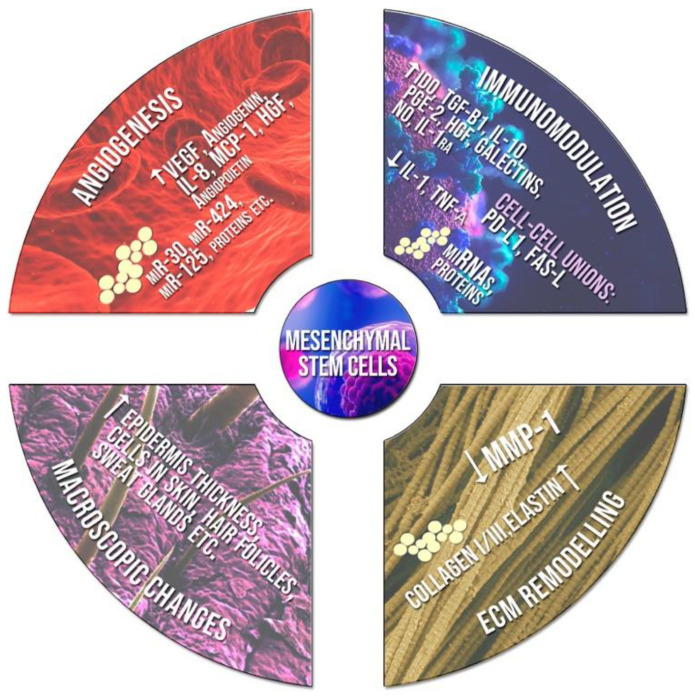
The effects of MScs in wound healing in terms of immunomodulation, ECM remodelling, macroscopic changes, and angiogenesis. Reproduced with permission from Heras et al. [118].

**Figure 9 polymers-14-05371-f009:**
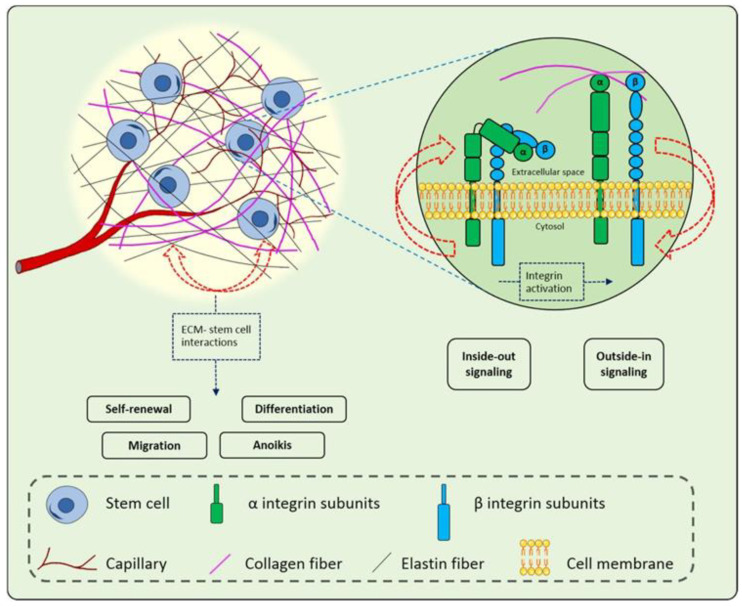
Graphical representation of the ECM-cell interactions. Integrins bind with collagen fibres, elastin, and laminin to bring about self-renewal, migration, differentiation, and anoikis of stem cell populations. Reproduced with permission from Rasouli et al. [128].

**Figure 10 polymers-14-05371-f010:**
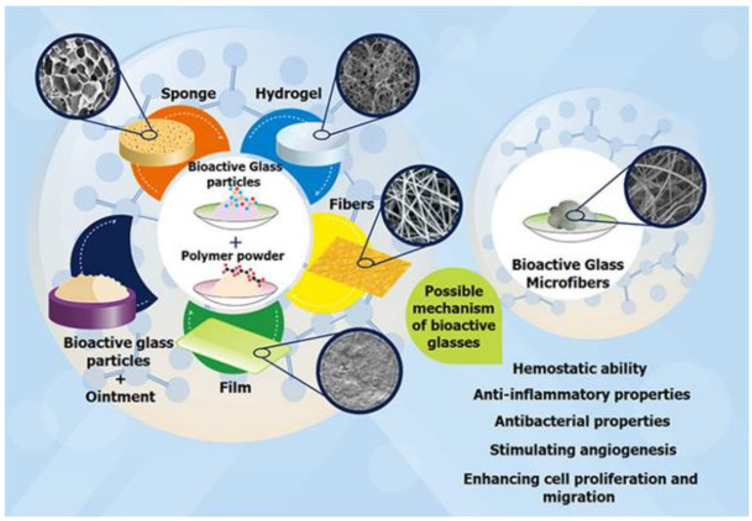
The different uses of bioactive glass in wound therapy and their mechanisms of action: (1) loading bioactive glass into ointments or polymeric-based matrices such as sponges, hydrogels, fibres or films; (2) microfibrous dressing loaded with bioactive glass. Reproduced with permission from Mehrabi et al. [148]. Copyright 2020 American Chemical Society.

**Table 1 polymers-14-05371-t001:** A concise summary of the common biopolymers highlighting their significant roles in wound healing.

Biopolymer	Composition	Biological Role in Wound Healing	Ref.
Chitosan	N-acetyl glucosamine linked by β-1, 4 glycosidic linkages	Haemostatic Induces fibroblast and keratinocytes migration and proliferation.	[28,30,74,75,76,77]
Alginate	1, 4-linked β-d-mannuronic and α-L-guluronic residues	Haemostatic Exudate draining Stimulated monocytes, induces fibroblast proliferation and migration	[39,41,42,78,79,80]
Collagen	Amino acids linked by amide linkage	Influences blood clotting cascade Induces fibroblast proliferation, induces ECM components, chemotactic for macrophages	[60,81]
Cellulose	β-d-Glucose linked by β-1, 4-glycosidic linkage	Antibacterial Retention of moisture, absorption of exudates	[37,44,82]
Hyaluronic acid	D-glucuronic acid and N-acetyl-d-glucosamine linked by β-1, 4 and β-1, 3 glycosidic linkages	Stimulates fibroblasts and keratinocytes proliferation and migration, anti-inflammatory	[53,54,55,56,83]

**Table 2 polymers-14-05371-t002:** A comprehensive summary of the aforementioned applications in wound healing.

Wound Dressing (with Application)	Biochemical Cue	Biological Performance	Ref.
In situ forming poly (L-lactic acid)-Pluronic L35 hydrogel loaded with antimicrobial peptides 57 nanoparticles (chronically infected wounds)	AMP nanoparticles	Promoted cutaneous wound healing by enhancing granulation tissue formation, increasing collagen deposition, and promoting angiogenesis	[92]
A composite hydrogel for the co-delivery of antimicrobial peptides and platelet-rich plasma (chronically infected wounds)	AMP with PRP	Improved wound healing in a diabetic mouse infection model by controlling inflammation, accelerating collagen deposition and angiogenesis	[93]
Collagen gels with CMP-immobilized or encapsulated DNA polyplexes (PDGF-BB delivery) (wound regeneration)	PEI DNA polyplexes with CMPs	Improved expression of PDGF-BB, proliferation, extracellular matrix production, and chemotaxis	[155]
Encapsulation of collagen mimetic peptide-tethered vancomycin liposomes in collagen-based scaffolds (chronic MRSA-infected wounds)	CMP	Sustained vancomycin release and enhanced in vitro and in vivo antibacterial properties against MRSA and closure rates	[110]
CMP-SubP conjugate (diabetic wounds)	CMP as a pylon with Substance P	CMP anneals to damaged collagen strands. Enhanced wound closure with noteworthy re-epithelialization and reduced inflammation in db/db mice	[106]
CMP-TGF-β-inducing peptide conjugate (severe wounds)	CMP as a pylon with peptide LTGKNFPMFHRN	Enhanced collagen deposition and wound closure in db/db mice by upregulation of the TGF-β signaling pathway	[117]
Mesenchymal stem cell spheroids embedded in an injectable thermosensitive semi-IPN hydrogel (cutaneous wounds)	MSCs	Faster wound closure in full-thickness wound models with reduced scarring. Well-organized collagen fibrils and high expression of the angiogenesis biomarker CD31 were also noted	[122]
Human MSCs in a PVA membrane (chronic wounds)	MSCs	Evaluated in dog non-healing skin lesions advancement in skin regeneration with a decreased expansion of ulcerated areas	[123]
Silk fibroin scaffold primed with adipose mesenchymal stromal cells (chronic diabetic ulcers)	MSCs	Improved tissue regeneration and reduction in wound region in db/db mice. Enhanced angiogenesis and matrix remodeling	[124]
Allogeneic Adipose-Derived Stem Cell-Hydrogel Complex (diabetic foot ulcers)	ASCs	Fifty-nine patients in a randomized clinical trial. Complete wound closure was achieved for 82% in the treatment group and 53% in the control group in week 12.	[126]
Autologous bone marrow nuclear cells (pressure ulcers)	BM-MNCs	In nineteen patients (86.36%), the pressure ulcers treated with BM-MNCs had fully healed after 21 days. Reduced hospital time, reduced treatment application time, and no reoccurrence of resolved ulcers was noted	[127]
Adipose-derived stem cells seeded on acellular dermal matrix grafts (full-thickness cutaneous wounds)	ASCs	Enhanced wound healing, angiogenesis, neo-vascularization, and VEGF-expressing ASCs were detected	[135]
Acellular dermal matrix with mesenchymal stem cell (full-thickness cutaneous wounds)	MSCs	Induced angiogenesis more efficiently than NPWT in rat models and improved neo-vascularization of the acellular dermal matrix	[136]
Platelet-Rich Plasma Based Dual-Network Hydrogel (various wound treatments)	PRP	In rats, the gel promoted rapid re-epithelialization, up-regulated growth factors, and early transitions in the wound healing and angiogenesis stages. It also exhibited superior healing efficiency in a porcine wound model.	[144]
Allogeneic Platelet-Rich Plasma Therapy (chronic wounds)	PRP	60-patient randomized clinical trial showed improved chronic wound healing	[145]
Platelet-rich plasma gel (diabetic foot ulcers)	PRP	Longitudinal and single-arm trial of 100 patients. The wound area significantly decreased, and healing times were reduced to 8 weeks	[146]
Thermosensitive bioglass/agarose–alginate composite hydrogel (chronic wounds)	BG	Enhanced vasculature and epithelium formation in a rabbit ear ischemic wound model	[153]
Bioglass-activated albumin hydrogel (chronic wounds)	BG	In the full-thickness excisional chronic wound model in mice, the gel stimulated angiogenesis, neo-vascularization, and enhanced epithelium regeneration	[154]
Biochemical and structural cues of 3D-printed matrix (MSC-based therapies)	MSCs	Biochemical and structural cues of 3D-printed matrix synergistically directed MSC differentiation to functional sweat glands in vitro and in vivo	[156]

AMP, antimicrobial peptide; ASC; adipose derived stem cells; CMP, collagen mimetic peptide; BG, bioglass; BM-MNC, bone marrow mononuclear cells; MSC, mesenchymal stem cells; PEI, polyethylenimine; PRP, platelet rich plasma.

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
