# Peer review of "Biopolymer-Based Wound Dressings with Biochemical Cues for Cell-Instructive Wound Repair"

_polymers, 2022, doi:10.3390/polym14245371_

Round 1
Reviewer 1 Report
The review entitled “A review on the design of polymer based wound dressings with biochemical cues for cell-instructive wound repair” presents a twenty-five-page manuscript with 99 references devoted to describing the recent advances in multifunctional biomacromolecule-based wound dressing materials. The authors divided the manuscript into two main sections: (1) an overview of the main biological events related to the wound healing process; (2) biochemical cues in wound healing applications. After reading the title of the manuscript, I do not think it reflects its entire content. The design of polymer-based wound dressing is not essentially covered. Additionally, many articles that address his topic (e.g., https://doi.org/10.1016/j.ijpharm.2021.120270; https://doi.org/10.1016/j.bioadv.2022.212738; https://doi.org/10.1021/acsabm.2c00035) were not mentioned. Furthermore, the authors did not justify the importance/novelty of the present review. Hence, I suggest a major revision of the manuscript for further consideration for publication.
Author Response
Thank you for these pertinent comments.
Please see attached for detailed responses.

Reviewer 2 Report
Recently, wound healing dressings based on functional polymer materials have attracted a lot of attention from scientists around the world, and this topic is very relevant. The review presented for review is a fairly detailed analysis of current trends in this field, and therefore has a great prospect - the introduction of completely new ideologies and approaches to the treatment of wound surfaces, including the study of molecular mechanisms and cellular control systems for the restoration of damaged tissue surfaces. And in this regard, without a doubt, the article is of great practical and scientific interest.. However, the manuscript needs to be finalized.
Some comments are:
1. Figure 2 already has a reference [21]. Why was the text reference added?
2. Chapter 3.1.1 should have added information on other antimicrobial peptides and hydrogels and additional references.
For example:
Malheiros, P. S. Jozala, A. F.; Pessoa-Jr., A.; Vila, M.M.D.C.; Balcão, V.M.; Franco, B.D.G.M. Immobilization of Antimicrobial Peptides from Lactobacillus Sakei Subsp. Sakei 2a in Bacterial Cellulose: Structural and Functional Stabilization. Food Packaging and Shelf Life. 2018, 17, 25–29.
Bayazidi, P.; Almasi, H.; Asl, A.K. Immobilization of lysozyme on bacterial cellulose nanofibers: Characteristics, antimicrobial activity and morphological properties. Int. J. Biol. Macromol. 2018, 107, 2544–2551.
3. Pg 14, line 596. What does AA mean in BG/AA composite? Need to clarify the abbreviation.
4. Pg 15, line 608. Similarly: What does HAS? Need to clarify the abbreviation.
5. The authors state in the chapter B 4. Further perspectives: «Moreover, biopolymers actively influence wound healing, thus selection of the biopolymer/s during synthesis is crucial in outlining biological performance. Biopolymers also offer biophysical structural prompts to direct cells for advantageous healing».
But in our opinion, the issue concerning the choice of biopolymers is not fully considered in the review. The number of considered biopolymers is limited.
6. Biochemical aspects are considered in detail. In addition, it would be interesting to expand the discussion of the biophysical aspects of the cellular microenvironment.
7. The issue under consideration regarding the rate of release of bioactive ions from the Bg/HAS hydrogel by the amount of BG is not entirely successful in its quantitative assessment.
8. The authors point out that bioactive glasses stimulate the secretion of GF (PDGF, EGF, TGFß, VEGF), but this is little discussed in the article
9. In our opinion, there are few illustrations in the article, while very interesting material is presented that suggests illustrative support.
Author Response
Thank you for these valuable comments.
Please see attached for detailed responses.

Reviewer 3 Report
This review introduced the wound healing dressings based on functionalized biopolymeric materials including peptide therapy, collagen matrices, stem cells, decellularized matrices, platelet rich plasma and biometals. As innovative wound dressings and skin substitutes are urgently need in the clinics right now, the review is of significant scientific importance. The topic is interesting. The langue is good but still needs improvement. Eg: Line 12-15 “... for hospital stay ” could be deleted. Line 77-78, the sentence is hard to understand. Line 111, Line 118, would these sentences be started at a separate paragraph?
Besides, why the author only chose these functional biomaterials for review? How about the polysaccharides, hyaluronic acid based hydrogels?
Author Response

(The authors gave the same response as above.)

Round 2
Reviewer 3 Report
The paper is OK for publication.